# Establishment of dry-chemistry-based reference intervals of routine liver function tests for the adult population of Gandaki Province, Nepal

**Asmita Sharma**[1]*, **Daya Ram Pokharel**[2], **Ganesh Dhakal**[1]

**1** School of Health and Allied Sciences, Pokhara University, Pokhara, Gandaki Province, Nepal,
**2** Department of Biochemistry, Manipal College of Medical Sciences, Pokhara, Gandaki Province, Nepal

* atimsa53@gmail.com

**Data Availability Statement:** The datasets generated and analyzed during the current study have been added to the Harvard Dataverse network

## Abstract

Every clinical laboratory should ideally establish its own population-specific reference intervals (RIs) to promote precision and evidence-based medicine. However, clinical laboratories in Nepal find it easier to follow external RIs than establish their own, leading to a lack of RIs specific to the local population. This study thus aimed to establish RIs of routine LFTs for the adult population of Gandaki Province, Nepal, and compare them with the current RIs used by our laboratory. We established the dry-chemistry-based reference intervals of 11 common LFT parameters for the adult population of Gandaki Province, Nepal using the direct priori-based method. The combined and sex-specific 95% double-sided RIs of total protein, albumin, globulin, A/G ratio, bilirubin, aspartate aminotransaminase (AST), alanine aminotransaminase (ALT), AST/ALT ratio, and alkaline phosphatase (ALP) were established using non-parametric percentile method. The new RIs were also compared with the currently used RIs that were adopted from the reagent kit inserts. The newly established RIs for each LFT were: Total proteins: 68.0–69.0g/L, albumin: 39.0–52.0g/L; globulin: 27.0–42.0g/L; A/G ratio: 1.1–1.8; total bilirubin: 5.13–25.65μmol/L (0.30–1.50mg/dl); unconjugated bilirubin: 1.71–17.10μmol/L (0.10–1.00mg/dl); conjugated bilirubin: 0.00–10.26 μmol/L (0.00–0.60mg/dl); AST: 20.0–43.2U/L; ALT: 11.0–53.0 U/L; AST/ALT ratio: 0.7–2.1; ALP: 42.0–135.4U/L. The RIs of albumin, globulin, A/G ratio, AST, ALT, and AST/ALT ratio differed significantly ($p < 0.05$) between males and females. Moreover, calculated out-of-range values showed that up to 4–40% of apparently healthy adults were classified as having abnormal test results based on current RIs. The newly established RIs fulfil the need for population and platform-specific RIs for the adult population of Gandaki Province of Nepal and bring more conformity and accuracy in interpreting the LFT results, diagnosis of hepatobiliary diseases, clinical decision-making, monitoring the success of therapy and future liver specific biomedical researches within the Gandaki Province of Nepal.

with the following details. https://doi.org/10.7910/DVN/CXPWL8.

**Funding:** AS got Masters Thesis Grant from the University Grants Commission Nepal (grant number (MRS-78-79-HS-05)) https://www.ugcnepal.edu.np and have met all the conditions. The funders had no role in study design, data collection and analysis, decision to publish, or preparation of the manuscript.

**Competing interests:** The authors have declared that no competing interests exist.

# Introduction

The liver is a vital organ that performs various functions, such as metabolism, detoxification, synthesis, and secretion of substances essential for maintaining homeostasis [1]. Liver function tests are a group of biochemical assays that measure the levels of various enzymes, proteins, and substances in the blood that reflect the health and functionality of the liver. They are widely used in clinical practice to diagnose and monitor various liver diseases, such as hepatitis, cirrhosis, jaundice, and liver cancer [2]. Liver function tests are also useful for assessing the effects of drugs, toxins, and alcohol on the liver, as well as evaluating the liver's ability to metabolize and excrete substances [3].

The interpretation of liver function tests relies on the comparison of the measured values with reference intervals, which are ranges of values within which a physiological measurement is considered normal for a healthy individual. Reference intervals are established by measuring the target parameter in a large sample of apparently healthy individuals from a specific population and calculating the mean ± 2 standard deviations, which encompasses 95% of the population. They are important decision-support tools for clinical diagnosis and management, as well as for screening and follow-up of subjects in clinical trials [4].

However, reference intervals are not universal and may vary depending on various factors, such as age, sex, ethnicity, diet, lifestyle, medications, and comorbidities. Therefore, national and international guidelines recommend that each laboratory establish its reference intervals based on its local population characteristics and analytical methods [5]. The use of reference intervals derived from dissimilar populations or methods may lead to misinterpretation of clinical findings and inappropriate interventions [6, 7].

In Nepal, most diagnostic laboratories use reference intervals provided by manufacturers or literature sources, which are mainly based on Western populations. However, these reference intervals may not be suitable for the Nepalese population, which has diverse genetic and environmental influences. Moreover, there is a lack of harmonization and standardization of analytical methods among laboratories in Nepal, which may affect the comparability and accuracy of results [8]. In particular, liquid chemistry-based methods are more commonly used than dry chemistry-based methods in Nepal due to their lower cost and higher availability. However, liquid chemistry-based methods require more sample volume and processing time than dry chemistry-based methods, which may affect the stability and quality of the analytes. Dry chemistry-based methods offer advantages such as faster turnaround time, less sample volume, and less interference from the hemolysis or lipemia [9]. Therefore, it is important to compare and validate the results obtained by different methods and establish method-specific reference intervals.

To address this gap in knowledge and practice, we conducted a study to establish dry-chemistry-based reference intervals for eight liver function test parameters (total protein, albumin (A), globulin (G), A/G ratio, total bilirubin, unconjugated bilirubin, conjugated bilirubin, aspartate aminotransferase (AST), alanine aminotransferase (ALT), AST/ALT ratio and alkaline phosphatase) in a representative sample of healthy adults from Gandaki Province in Nepal. We also compared these reference intervals with other dry chemistry-based RIs obtained from reagent kit inserts and published literature sources. We followed the most widely followed guidelines (C28-A3c) published by the Clinical and Laboratory Standards Institute (CLSI) and the International Federation for Clinical Chemistry and Committee on Reference Intervals and Decision Limits (IFCC-CRIDL) for the selection of reference individuals and the calculation of reference intervals [5]. Our study aims to provide the first reliable and relevant reference intervals for liver function tests that can improve the quality and accuracy of clinical diagnosis and management of liver diseases in the Gandaki Province of Nepal.

## Materials and method

### Study design

This was a population-based cross-sectional study conducted at the Manipal Teaching Hospital (MTH), Pokhara, Nepal from April 2022 to September 2022. This study was designed as per the recent CLSI C28-A3 guidelines set forth by the Clinical and Laboratory Standards Institute (CLSI) and the International Federation for Clinical Chemistry and Committee on Reference Intervals and Decision Limits (IFCC-CRIDL) [5]. The collected blood samples were processed and analyzed at the Department of Clinical Biochemistry, Manipal Teaching Hospital, Pokhara, Nepal.

### Sample size, period, and area

The sample size of this study was determined according to the IFCC-CSLI EP28-A3c guidelines [5]. According to this guideline, a minimum of 120 study samples by partition is required to establish the reference intervals using conventional statistical methods. As we aimed to partition our study parameters for both males and females (two partitions), our minimum sample size was 120x2 = 240. Hence, keeping in mind the margin, we included about 13% extra participants in each partition, making the final total sample size 136x2 = 272. A priori convenience sampling method was employed for the selection of study participants.

The study participants for this study were residents of the Gandaki Province of Nepal representing all its 11 districts (Fig 1) who visited MTH as attendees of their admitted patients during the study period. The Gandaki Province extends from 28.35˚N to 84.05˚E, with a total area of 21856 km², and a total population of 2,466,427 [10].

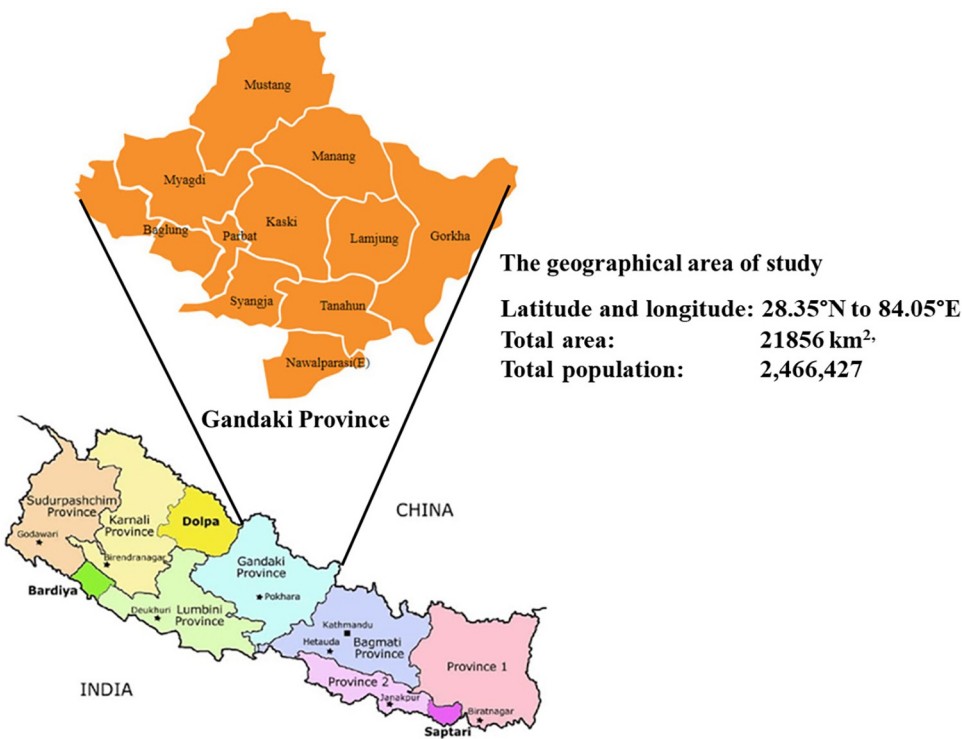

**Fig 1. The political map of Nepal and its Gandaki Province showing the area of study.** (https://www.researchgate.net/publication/352155539_Marginal_gains_borderland_dynamics_political_settlements_and_shifting_centre-periphery_relations_in_post-war_Nepal/figures?lo=1).

**Table 1. Inclusion and exclusion criteria for reference population selection.**

| Inclusion Criteria | Exclusion Criteria |
|---|---|
| BMI between 18.5–27.5 kg/m$^2$. | Presence of acute or chronic infections |
| | History of kidney or digestive diseases |
| | Presence of metabolic or nutritional disorders |
| | Hospitalization or serious illness within the preceding 4 weeks |
| | Use of replacement or supplement therapies such as thyroxine or insulin, |
| | Habit of excessive alcohol or tobacco consumption |
| | Massive blood loss |
| | Female participants who were Pregnant, breastfeeding, or within 6 months post-childbirth |
| | BMI < 18.5 or > 27.5 kg/m$^2$ |

## Reference population and selection criteria

The reference population for this study consisted of apparently healthy adults of 18–70 years who resided in one of the 11 districts of the Gandaki province of Nepal. Eligible participants were selected after a general screening using a set of medical history-centered questionnaires, general physical examination, and certain screening blood tests for chronic metabolic and infectious diseases (**Table 1**).

## Data collection for baseline variables

A structured and pre-validated questionnaire was administered to the study participants within the premises of Manipal Teaching Hospital. Data encompassing age, sex, disease history, family medical history, race, dietary habits, physical activity levels, smoking, and alcohol consumption patterns were meticulously recorded. Alcoholics were defined as participants who reported drinking ethyl alcohol on an occasional basis but not regularly, while non-alcoholics were defined as participants who had never consumed alcohol or had abstained for at least a year. Further, anthropometric measurements including height, weight, hip and waist circumferences, waist-hip ratio, and BMI were acquired using established protocols. Blood pressure was also measured using a mercury sphygmomanometer.

## Sample collection processing and storage

Participants were advised to keep their routine diet and daily activities and avoid smoking or drinking habits for 2–3 days before being called to the sample collection center of MTH. On the day of sample collection, about 5 ml of venous blood samples were drawn from each consented participant and collected in gel tubes containing clot activators. Subsequently, the samples were allowed to clot at room temperature for about 20 to 30 minutes and then centrifuged at 4000 rpm for 10 minutes. Samples exhibiting lipemia, hemolysis, or icterus were excluded from further analysis. Serum samples were either analyzed immediately or stored at -20°C until analysis.

## Instrumentation and analytical methods

A state-of-the-art fully automated dry chemistry-based analyzer (VITROS 4600 Chemistry system, Ortho Clinical Diagnostics, UK) connected with a bidirectional laboratory information system, was used for the determination of screening parameters (serum glucose and creatinine) and LFT parameters (serum total proteins, albumin, globulin, total, direct and indirect

**Table 2. Principle of the analytes and traceability of the results.**

| Parameter | Method | Traceability |
|---|---|---|
| Total Protein | Biuret | Certified NIST and SRM 927 [12] |
| Albumin | Bromo Cresol Green | Certified NIST and Total Protein SRM 927 [13]. |
| Total Bilirubin | Diazo | Certified NIST, SRM 916 to calibrate the NCCLS credentialed Jendrassik-Grof method [14]. |
| Direct and Indirect Bilirubin | Mordant | High-Performance Liquid Chromatography (HPLC) method [15]. |
| AST | IFCC with PLP | Aspartate aminotransferase method recommended by IFCC, adapted to a centrifugal analyzer at 37˚C [16]. |
| ALT | IFCC with PLP | Alanine aminotransferase method recommended by IFCC [17]. |
| ALP | AMP | Alkaline phosphatase method recommended by IFCC, adapted to an automated analyzer at 37˚C [18]. |

**Abbreviations used: NIST:** National Institute of Standards and Technology, **SRM:** Standard Reference Materials, **NCCLS:** National Committee for Clinical Laboratory Standards, **HPLC:** High-Performance Liquid Chromatography, **AST:** Aspartate aminotransferase, **ALT**: Alanine aminotransferase, **ALP**: Alkaline phosphatase, **AMP:** Adenosine monophosphate, **IFCC:** International Federation of Clinical Chemistry, **PLP:** Pyridoxal 5'-phosphate

bilirubin, aspartate aminotransferase (AST), alanine aminotransferase (ALT), and alkaline phosphatase (ALP)) following standardized protocols provided by the reagent manufacturer. The assay methods used and their traceability for each analyte are presented in **Table 2**. The serum creatinine-based glomerular filtration rate (eGFRcr) was estimated using an equation described by chronic kidney disease epidemiology collaboration (CKD-EPI) [11].

## Quality control and assurance

Stringent quality control procedures were enforced throughout the study. Calibration of instruments was performed using calibrators and internal controls. To establish a valid and reliable reference interval for each LFT parameter, daily internal quality control procedures and the same lots of reagents were used throughout this study (S1 Table).

Quality control sera at two levels were utilized for each analyte. The coefficients of variation (CV%) for inter-assay (reproducibility) and intra-assay (repeatability) were assessed and were all within acceptable ranges. Inter-assay CV% for serum total protein, albumin, total bilirubin, unconjugated bilirubin, conjugated bilirubin, AST, ALT, ALP, glucose, creatinine, were 0.64, 1.49, 0.39, 0.35, 0.89, 2, 2, 2, 0.9, 0.53 respectively. Similarly, intra-assay CV% for these same parameters were 1.34, 0.66, 1.82, 1.32, 0.33, 0.39, 0.19, 0.27, and 1.30, respectively. The daily internal QC values for these parameters were within ±2 SD of the target values respectively (S2 Table) Monthly participation in the External Quality Assessment Schemes (EQAS) administered by the Christian Medical College, Vellore further ensured the precision and reliability of our laboratory results.

## Ethical consideration

Prior approval (MEMG/IRC/520/GA) for this study was obtained from the Institutional Research and Ethics Committee of the Manipal Teaching Hospital, Pokhara, Nepal. The aims and objectives of this study were clearly explained to the study subjects by the investigators and written informed consent was obtained from each of them before collecting their venous blood samples and data on their baseline variables.

## Statistical analysis

Data management and statistical analyses were conducted using MS Excel, IBM SPSS (version 20), and MedCalc software (version 17.2). Descriptive statistics was used to summarize the demographic, anthropometric, and lifestyle data. The Gaussian distribution of the test variables was assessed using D'Agostino-Pearson Omnibus, Kolmogorov-Smirnov, and Shapiro-Wilk tests. The outlier data were identified and removed using Box plots, suggested by Horn and Pesce and Tukey's rule. Since most of the data were distributed non-normally, non-parametric Mann-Whitney U and Chi-square tests were used to compare quantitative and qualitative variables, respectively. The total and sex-stratified 95% double-sided RIs were established using a priori approach as recommended by CLSI-IFCC C28-A3 guidelines. The mean, median, 2.5$^{th}$ and 97.5$^{th}$ percentiles and range of RIs were determined. The conformity between new RIs and the old adopted RIs was checked by calculating the out-of-range (OOR) percentage. Statistical significance was set at $p < 0.05$.

## Results

### Characteristics of study participants

There was a total of 272 enrolled study participants who represented all 11 districts of the Gandaki Province of Nepal (S3 Table). There were 50% (n = 136) males and 50% (n = 136) females, with a mean total age of 37.8±15.0 years Males and females differed significantly (p<0.05) in terms of their alcohol intake habit, status of hepatitis B vaccination, exercise pattern, height, weight, waist-to-hip ratio, systolic and diastolic blood pressure, and serum creatinine values (Table 3).

### Distribution patterns of LFT parameters

Histograms with distribution curves were plotted for analysis of the distribution pattern of all liver function test parameters and presented in Fig 2. Except for total protein and albumin, all other LFT parameters were distributed non-normally.

### The linear monotonic association between LFT parameters and baseline covariates

The total protein level and alkaline phosphatase level were significantly correlated (p<0.05) only with age while albumin and A/G ratio were significantly correlated (p<0.05) with both age and sex. Bilirubin levels, besides being significantly correlated with age and sex, also showed additional correlation with anthropometric measures, blood glucose level, creatinine level, eGFRcr, and all other LFT parameters. Albumin also showed an additional correlation (p<0.05) with WHR, BMI, serum glucose, creatinine, eGFRcr, and total protein.

The strength of the association between the variables is represented by the color intensity of the square boxes as shown in Fig 3. The red color denotes a positive correlation while the blue color denotes a negative correlation. The correlation analysis was carried out using the Spearman rho test. The significant associations are flagged with *. *p<0.05, **p<0.001.

### Establishment of combined and sex-specific RIs of LFT parameters

The mean, median, and 95% RIs with lower and upper limits and intervals for the routine LFT parameters are presented in Table 4. The combined 95% RIs for each parameter, respectively were: total protein: 6.8–8.9g/dL; albumin: 3.9–5.2 g/dL; globulin: 2.7–4.2g/dL; A/G ratio: 1.0–1.8; total bilirubin: 5.13–25.65μmol/L; unconjugated bilirubin: 1.71–17.10 μmol/L; conjugated

**Table 3. Sociodemographic characteristics of study participants.**

| Characteristics | Female (n = 136) | Male (n = 136) | P-value | Total (n = 272) |
|---|---|---|---|---|
| **Age (years)** | 38.4 ± 14.5 | 37.2 ± 15.5 | 0.527 | 37.8±15.0 |
| Age group (years) | | | | |
| 18–29 | 47 (17.3) | 59 (21.7) | 0.088 | 106 (39) |
| 30–39 | 27 (9.9) | 29 (10.7) | | 56 (20.6) |
| 40–49 | 29 (10.7) | 13 (4.8) | | 42 (15.4) |
| 50–59 | 18 (6.6) | 16 (5.9) | | 34 (12.5) |
| 60–69 | 15 (5.5) | 19 (7) | | 34 (12.5) |
| **Diet** | | | 0.652 | |
| Vegetarian | 3 (1.1) | 2 (0.70) | | 5 (1.8) |
| Non-vegetarian | 133 (48.9) | 134 (49.3) | | 267 (98.2) |
| **Alcohol** | | | <0.001 | |
| Yes | 8 (2.9) | 54 (19.9) | | 62 (22.8) |
| No | 128 (47.1) | 82 (30.1) | | 210 (77.2) |
| **Smoking** | | | 0.051 | |
| Yes | 5 (1.8) | 13 (6.6) | | 18 (6.6) |
| No | 131 (48.2) | 123 (45.2) | | 254 (93.4) |
| **Hep B vaccination** | | | 0.005 | |
| Yes | 5 (1.8) | 18 (6.6) | | 23 (8.5) |
| No | 131 (48.2) | 118 (43.4) | | 249(91.5) |
| **Exercise** | | | <0.001 | |
| Low | 126 (46.4) | 97 (35.7) | | 223 (82) |
| Medium | 8 (2.9) | 28 (10.3) | | 36 (13.2) |
| High | 2 (0.7) | 11 (8.1) | | 13 (4.8) |
| **Mensuration** | | | | |
| Regular | 84 (30.9) | - | | 84 (30.9) |
| Irregular | 20 (7.4) | - | | 20 (7.4) |
| Menopause | 32 (11.8) | - | | 32 (11.8) |
| **Blood pressure** | | | | |
| SBP (mmHg) | 111.5 ± 10.1 | 117.9 ± 6.9 | <0.001 | 114.7 ± 9.2 |
| DBP (mmHg) | 73.8 ± 7.6 | 78.3 ± 5.7 | <0.001 | 76.1 ± 7.0 |
| **Anthropometry** | | | | |
| BMI (kg/m$^2$) | 24.1 ± 2.8 | 24.0 ± 2.3 | 0.933 | 24.1±2.6 |
| Normal weight | 48 (17.6) | 45 (16.5) | 0.048 | 93 (34.2) |
| Overweight | 26 (9.6) | 43 (15.8) | | 69 (25.4) |
| Obese | 62 (22.8) | 48 (17.6) | | 110 (40.4) |
| W/H ratio | 0.9 ± 0.1 | 0.9 ± 0.1 | <0.001 | 0.9 ± 0.1 |
| **Metabolic parameter** | | | | |
| Glucose (mmol/L) | 5.3±0.5 | 5.2 ± 0.5 | 0.222 | 5.2 ± 0.5 |
| **Renal function tests** | | | | |
| Creatinine (μmol/L) | 61.9 ± 8.8 | 79.6 ± 17.7 | <0.001 | 70.7 ± 17.7 |
| eGFR$_{cr}$ (ml/min/1.73 m$^2$) | 106.2 ± 18.5 | 106.8 ± 18.4 | 0.787 | 106.5 ± 18.4 |

Abbreviation: **Hep B**: Hepatitis B virus, **SBP**: Systolic Blood Pressure, **DBP**: Diastolic Blood Pressure, **BMI**: Body Mass Index, **W/H ratio**: Waist-to-Hip ratio, **eGFRcr**: Serum creatinine-based estimated glomerular filtration rate. The continuous variables are presented as mean ± SD and categorical variables are presented as numbers and percentages [n (%)]. Chi-squared test was used to compare the differences between categorical variables while the Mann-Whitney U test was used for the continuous variables. p-values were considered statistically significant when <0.05.

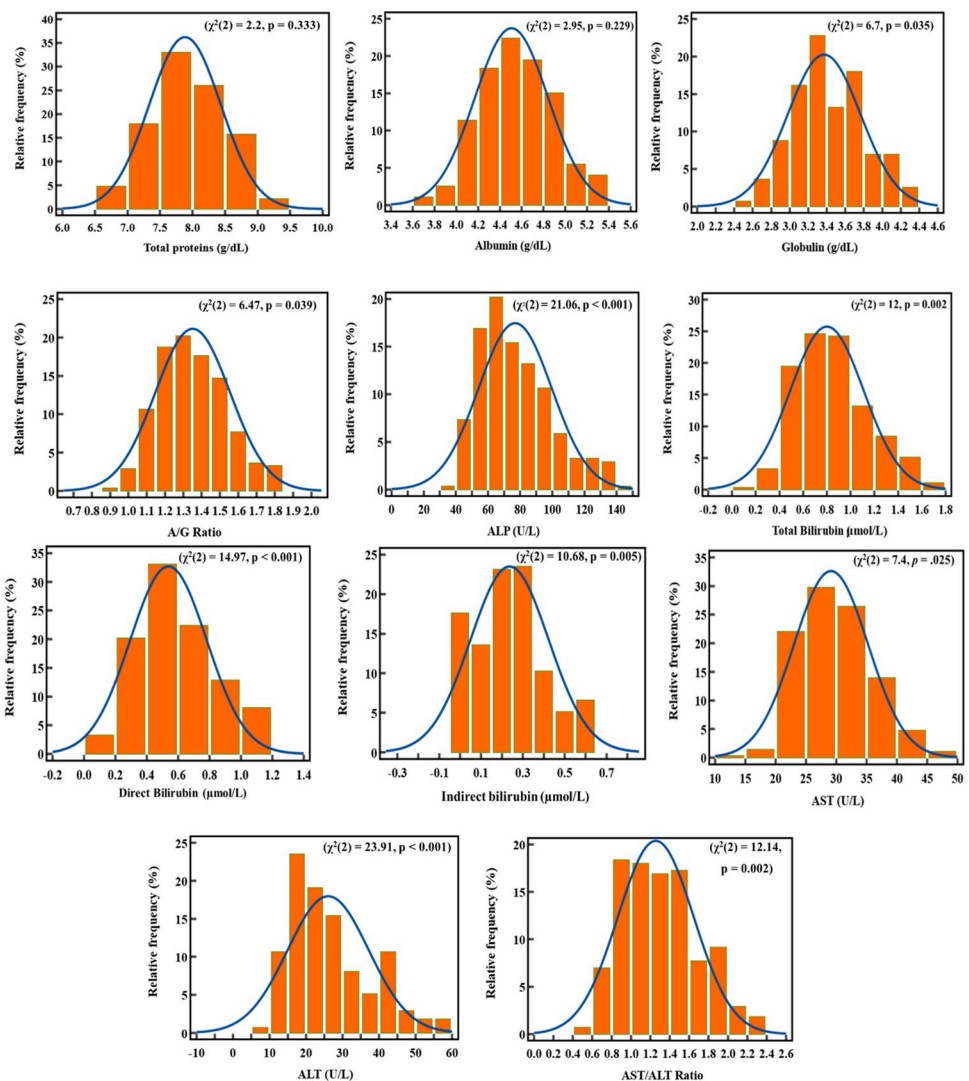

**Fig 2. The distribution pattern of the liver function test parameters in the study participants observed (orange boxes) versus standard (blue curves).**

bilirubin: 0.00–10.26 μmol/L; AST: 20.0–43.2U/L; ALT: 11.0–53.0 U/L; AST/ALT ratio: 0.7–2.1; ALP: 42.0–135.4U/L. The 95% RIs for males and females, respectively, were: total protein: males (M): 6.9–9.2g/L; females (F): 6.7–8.9g/dL; albumin: M: 4.0–5.3g/dL; F: 3.7–5.1g/dL; globulin: 2.6–4.1g/dL and 2.7–4.0g/dL; A/G ratio: 1.1–1.8 and 1.0–1.7; total bilirubin: 4.10–26.68 μmol/L and 5.13–23.94 μmol/L; unconjugated bilirubin: 1.71–17.10 μmol/L and 1.71–17.10 μmol/L; conjugated bilirubin: 0.00–10.26 μmol/L and 0.00–10.26 μmol/L; AST: 20.0–43.6 U/L and 19.4–43.6 U/L; ALT: 11.4–53.0 U/L and 11.0–44.0 U/L; AST/ALT ratio: 0.7–1.8 and 0.8–2.1; ALP: 42.0–134.0U/L and 41.3–136.1 U/L. Among these, only RIs of albumin, globulin, A/G ratio, total bilirubin, unconjugated bilirubin, and conjugated bilirubin were found to differ significantly ($p < 0.05$) between male and female subjects.

## Comparison between new and old RIs

The out-of-range (OOR) proportions were calculated only to compare the new RIs determined by us and the currently used RIs in our laboratory. The total proportion of OOR values

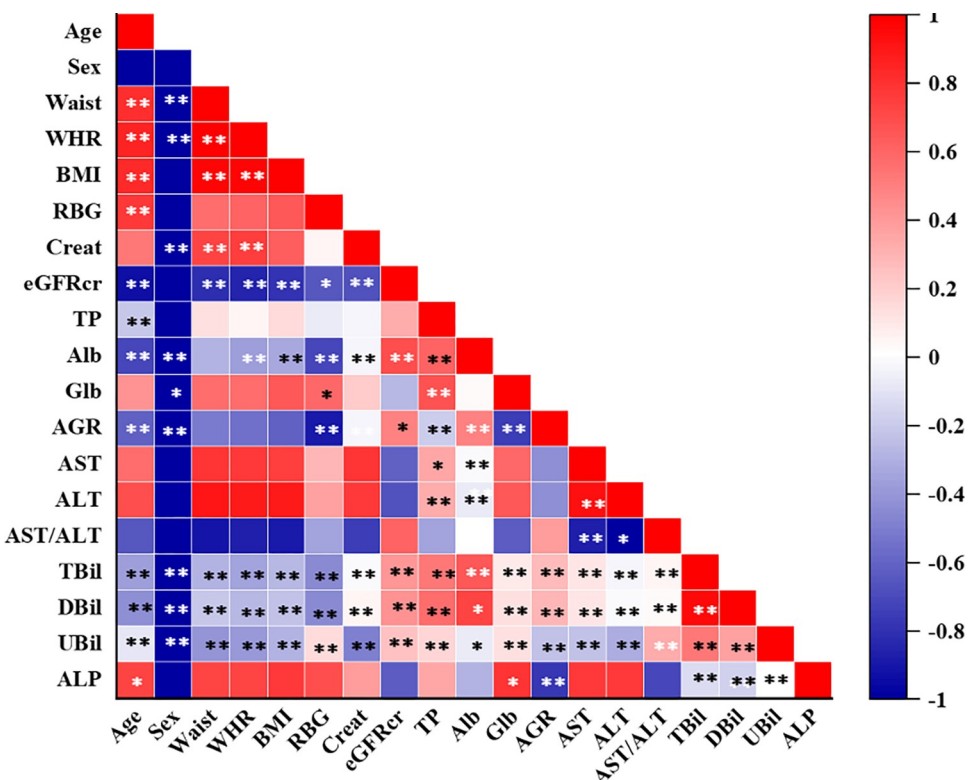

**Fig 3. Correlation matrix showing a linear monotonic association between LFT parameters and other baseline covariates.**

between these new and old sets of RIs was between 1–35%. There was a complete match between the existing and new RIs for unconjugated bilirubin. The OOR% was minimum (1.0%) for the AST and maximum for the AST/ALT ratio (39.6%). The OOR proportion values were ≥ 20% for total protein, globulin, A/G ratio, and conjugated bilirubin. The details of the comparison between new and existing RIs are presented in **Table 5**.

## Discussion

This study marks a significant milestone in the field of clinical biochemistry in Nepal, being the first to report both combined and sex-based Reference Intervals (RIs) for 11 common Liver Function Tests (LFTs) focusing on the general adult population of Gandaki province, Nepal. The necessity of such a study for the Nepalese population cannot be overstated, given the unique genetic, dietary, geographical and environmental factors that can influence these parameters. Moreover, such RIs for the local Nepalese population were never established before by any other clinical laboratories.

The main outcome of our study was the establishment of new dry-chemistry-based RIs of 11 LFTs using a priori method for both adult male and female populations of Gandaki Province, Nepal. Gandaki province, which lies in the central part of Nepal that adjoins Tibet's autonomous region of China in the North and India in the South, is the habitat of people with diverse races, ethnicities, dietary habits, environment and geography. Since, the whole terrain of Nepal is divided into plane, hills and high-altitude mountains from east to west, the population of Gandaki province represent almost 8.5% of the total population of Nepal [10]. These RIs were established using the highly recommended Clinical and Laboratory Standards

**Table 4. Reference Intervals (RIs) of common LFT parameters for the adult population of Gandaki Province, Nepal.**

| Tests | Unit | Sex | N | Mean ± SD | Median | Outliers | p-value | 95% Reference Interval, double-sided | | Interval | RI |
|---|---|---|---|---|---|---|---|---|---|---|---|
| | | | | | | | | 2.5th (90%CI) | 97.5th (90%CI) | | |
| TP | g/dl | T | 272 | 7.9 ± 6.0 | 7.9 | 0 | | 6.8 (6.7–6.9) | 8.9 (8.8–9.2) | 2.1 | 6.8–8.9 |
| | g/L | | | 79.0 ± 60.0 | 79 | | | 68 .0 (67.0–69.0) | 89.0 (88.0–92.0) | 21 | 68.0–89.0 |
| | g/dL | M | 136 | 7.9 ± 0.7 | 7.8 | 0 | 0.272 | 6.9 (6.8–7.1) | 9.2 (8.8–9.4) | 2.3 | 6.9–9.2 |
| | g/L | | | 79.0 ± 70.0 | 78 | | | 69.0 (68.0–71.0) | 92.0 (88.0–94.0) | 23 | 69.0–92.0 |
| | g/dl | F | 136 | 7.9 ± 0.8 | 7.9 | 0 | | 6.7 (6.5–6.9) | 8.9 (8.7–9.0) | 2.2 | 6.7–8.9 |
| | g/L | | | 79.0 ± 80.0 | 79 | | | 67.0 (65.0–69.0) | 89.0 (87.0–90.0) | 22 | 67.0–89.0 |
| ALB | g/dl | T | 272 | 4.5 ± 0.3 | 4.5 | 0 | | 3.9 (3.7–4.0) | 5.2 (5.1–5.3) | 1.3 | 3.9–5.2 |
| | g/L | | | 45.0 ± 30.0 | 45 | | | 39.0 (37.0–40.0) | 52.0 (51.0–53.0) | 13 | 39.0–52.0 |
| | g/dl | M | 136 | 4.6 ± 0.3 | 4.6 | 0 | <0.001 | 4.0 (3.9–4.1) | 5.3 (5.2–5.3) | 1.3 | 4.0–5.3 |
| | g/L | | | 46.0 ± 30.0 | 46 | | | 40.0 (39.0–41.0) | 53.0 (52.0–53.0) | 13 | 40.0–53.0 |
| | g/dl | F | 136 | 4.4 ± 0.3 | 4.4 | 0 | | 3.7 (3.7–3.9) | 5.1 (4.9–5.2) | 1.4 | 3.7–5.1 |
| | g/L | | | 44.0 ± 30.0 | 44 | | | 37.0 (37.0–39.0) | 51.0 (49.0–52.0) | 14 | 37.0–51.0 |
| GLOB | g/dl | T | 272 | 3.4 ± 0.4 | 3.3 | 0 | | 2.7 (2.6–2.7) | 4.2 (4.1–4.2) | 1.5 | 2.7–4.2 |
| | g/L | | | 34.0 ± 40.0 | 33 | | | 27.0 (26.0–27.0) | 42.0 (41.0–42.0) | 15 | 27.0–42.0 |
| | g/dl | M | 136 | 3.3 ± 0.4 | 3.3 | 0 | 0.012 | 2.6 (2.4–2.7) | 4.1 (4.0–4.2) | 1.5 | 2.6–4.1 |
| | g/L | | | 33.0 ± 40.0 | 33 | | | 26.0 (24.0–27.0) | 41.0 (40.0–42.0) | 15 | 26.0–41.0 |
| | g/dl | F | 136 | 3.4 ± 0.4 | 3.5 | 0 | | 2.7 (2.7–2.8) | 4.2 (4.1–4.3) | 1.5 | 2.7–4.2 |
| | g/L | | | 34.0 ± 40.0 | 35 | | | 27.0 (27.0–28.0) | 42.0 (41.0–43.0) | 15 | 27.0–42.0 |
| A/G | ratio | T | 272 | 1.4 ± 0.2 | 1.3 | 0 | | 1.0 (1.0–1.1) | 1.8 (1.7–1.8) | 0.8 | 1.0–1.8 |
| | | M | 136 | 1.4 ± 0.2 | 1.4 | 0 | <0.001 | 1.1 (1.1–1.1) | 1.8 (1.8–1.8) | 0.7 | 1.1–1.8 |
| | | F | 136 | 1.3 ± 0.2 | 1.3 | 0 | | 1.0 (0.9–1.0) | 1.7 (1.5–1.8) | 0.7 | 1.0–1.7 |
| TBIL | mg/dl | T | 272 | 0.8 ± 0.31 | 0.8 | 0 | | 0.30 (0.20–0.40) | 1.50 (1.50–1.60) | 1.2 | 0.30–1.50 |
| | μmol/L | | | 13.68 ± 5.30 | 13.68 | | | 5.13 (3.42–6.84) | 25.65 (25.65–27.36) | 20.52 | 5.13–25.65 |
| | mg/dl | M | 136 | 0.83 ± 0.32 | 0.8 | 0 | <0.001 | 0.24(0.10–0.40) | 1.56 (1.50–1.60) | 1.32 | 0.24–1.56 |
| | μmol/L | | | 14.19±5.47 | 13.68 | | | 4.10 (1.71–6.84) | 26.68 (25.65–27.36) | 22.58 | 4.10–26.68 |
| | mg/dl | F | 136 | 0.78 ± 0.29 | 0.7 | 0 | | 0.30 (0.30–0.40) | 1.40 (1.30–1.50) | 1.1 | 0.30–1.40 |
| | μmol/L | | | 13.33±4.95 | 11.97 | | | 5.13 (5.13–6.84) | 23.94 (22.23–25.65) | 5.81 | 5.13–23.94 |
| UBil | mg/dl | T | 272 | 0.54 ± 0.24 | 0.5 | 0 | | 0.10 (0.10–0.20) | 1.00 (1.00–1.10) | 0.9 | 0.10–1.00 |
| | μmol/L | | | 9.23 ± 4.10 | 8.55 | | | 1.71 (1.71–3.42) | 17.10 (17.10–18.81) | 15.39 | 1.71–17.10 |
| | mg/dl | M | 136 | 0.56 ± 0.25 | 0.5 | 0 | <0.001 | 0.10 (0.10–0.20) | 1.00 (1.00–1.10) | 0.9 | 0.10–1.00 |
| | μmol/L | | | 9.58± 4.28 | 8.55 | | | 1.71 (1.71–3.42) | 17.10 (17.10–18.81) | 15.39 | 1.71–17.10 |
| | mg/dl | F | 136 | 0.52 ± 0.24 | 0.5 | 0 | | 0.10 (0.10–0.20) | 1.00 (1.00–1.10) | 0.9 | 0.10–1.00 |
| | μmol/L | | | 8.89 ± 4.10 | 8.55 | | | 1.71 (1.71–3.42) | 17.10 (17.10–18.81) | 15.39 | 1.71–17.10 |

(*Continued*)

**Table 4.** (Continued)

| Tests | Unit | Sex | N | Mean ± SD | Median | Outliers | p-value | 95% Reference Interval, double-sided | | Interval | RI |
|---|---|---|---|---|---|---|---|---|---|---|---|
| | | | | | | | | 2.5th (90%CI) | 97.5th (90%CI) | | |
| CBIL | mg/dl | T | 272 | 0.24 ± 0.17 | 0.2 | 0 | | 0.00 (0.00–0.00) | 0.6 (0.60–0.60) | 0.6 | 0.00–0.6 |
| | µmol/L | | | 4.10 ± 2.91 | 0.68 | | | 0.00 (0.00–0.00) | 10.26 (10.26–10.26) | 10.26 | 0.00–10.26 |
| | mg/dl | M | 136 | 0.24 ± 0.18 | 0.2 | 0 | <0.001 | 0.00 (0.00–0.00) | 0.6 (0.60–0.60) | 0.6 | 0.00–0.6 |
| | µmol/L | | | 4.10 ± 3.08 | 0.68 | | | 0.00 (0.00–0.00) | 10.26 (10.26–10.26) | 10.26 | 0.00–10.26 |
| | mg/dl | F | 136 | 0.24 ± 0.16 | 0.2 | 0 | | 0.00 (0.00–0.00) | 0.6 (0.50–0.60) | 0.6 | 0.00–0.6 |
| | µmol/L | | | 4.10 ± 2.74 | 0.68 | | | 0.00 (0.00–0.00) | 10.26 (8.55–10.26) | 10.26 | 0.00–10.26 |
| AST | U/L | T | 272 | 29.1 ± 6.1 | 29 | 0 | | 20.0(18.0–20.0) | 43.2 (41.0–45.0) | 23.2 | 20.0–43.2 |
| | | M | 136 | 30.4 ± 6.2 | 30 | 0 | 0.217 | 20.0 (15.0–21.0) | 43.6 (40.0–45.0) | 23.6 | 20.0–43.6 |
| | | F | 136 | 27.9 ± 5.8 | 27 | 0 | | 19.4 (13.0–20.0) | 43.6 (38.0–45.0) | 24.2 | 19.4–43.6 |
| ALT | U/L | T | 269 | 26.0 ± 11.0 | 24 | 3 | | 11.0 (10.0–12.0) | 53.0 (48.0–57.0) | 42 | 11.0–53.0 |
| | | M | 136 | 28.8 ± 11.1 | 27 | 0 | 0.251 | 11.4 (9.0–13.0) | 53.0 (47.0–57.0) | 41.6 | 11.4–53.0 |
| | | F | 133 | 22.5 ± 8.9 | 20 | 3 | | 11.0 (9.0–12.0) | 55.0 (41.0–57.0) | 44 | 11.0–55.0 |
| AST/ALT | ratio | T | 265 | 1.2 ± 0.4 | 1.3 | 7 | | 0.7 (0.6–0.7) | 2.1 (2.0–2.2) | 1.4 | 0.7–2.1 |
| | | M | 132 | 1.1 ± 0.3 | 1.1 | 4 | 0.685 | 0.7 (0.6–0.7) | 2.1 (1.8–2.3) | 1.4 | 0.7–2.1 |
| | | F | 133 | 1.4 ± 0.4 | 1.3 | 3 | | 0.7 (0.4–0.8) | 2.1 (2.0–2.2) | 1.4 | 0.7–2.1 |
| ALP | U/L | T | 272 | 76.7 ± 22.6 | 73 | 0 | | 42.0 (41.0–44.0) | 135.4 (123.0–139.0) | 93.4 | 42.0–135.4 |
| | | M | 136 | 77.7 ± 23.9 | 73 | 0 | 0.765 | 42.0 (42.0–44.0) | 134.0 (121.0–140.0) | 92 | 42.0–134.0 |
| | | F | 136 | 76.0 ± 21.7 | 73 | 0 | | 41.4(36.0–46.0) | 136.1 (119.0–139.0) | 94.7 | 41.4–136.1 |

Abbreviation: **TP**: Total Protein, **ALB**: Albumin, **GLOB**: Globulin, **A/G**: Albumin/Globulin Ratio, **TBIL**: Total Bilirubin, **UBIL**: Unconjugated Bilirubin, **CBIL**: Conjugated Bilirubin, **AST**: Aspartate Transaminase, **ALT**: Alanine Transaminase, **ALP**: Alkaline Phosphatase, **LL**: Lower Limit, **UL**: Upper Limit, **CI**: Confidence Interval, **N**: Sample size, **RI**: Reference interval.

Institute and International Federation of Clinical Chemistry (CLSI-IFCC) guidelines (C28-A3c), ensuring their robustness and reliability [5]. The adherence to both internal and external quality assurance measures, and the use of fully automated random-access clinical chemistry analyzer, and traceable assay methods further strengthen the validity of our findings.

In the absence of locally established reference intervals, diagnostic laboratories in Nepal have to fully rely on external RIs provided in the kit insert or clinical chemistry textbooks. However, the use of such RIs is always questionable as they do not represent the local population and actual methods or platforms being used during the analytical measurements. Such irrational use of imported RIs may thus increase the risk of misinterpretation of test results and lead to wrong diagnosis of the diseases. This is the reason why every diagnostic laboratory is supposed to establish its own reference intervals for the population under its service. We chose OCD Vitros-based dry chemistry platforms because there is a growing trend of adopting these platforms in Nepal and also the significant lack of reliable published RIs specific to this

**Table 5. Comparison of the current reference interval established in this study with adopted reference interval analytes.**

| Analytes | Unit | Current RIs | Unit | New RIs | OOR, N (%) |
|---|---|---|---|---|---|
| TP | g/dL | 6.3–8.2 | g/dl | 6.8–8.9 | 78 (29%) |
| ALB | g/dL | 3.5–5.0 | g/dl | 3.9–5.2 | 18 (7%) |
| GLOB | g/dL | 2.0–3.5 | g/dl | 2.7–4.2 | 94 (35%) |
| A/G | Ratio | 1.2–2.5 | ratio | 1.0–1.8 | 55 (20%) |
| TBIL | mg/dL | 0.2–1.3 | mg/dl | 0.3–1.5 | 18 (7%) |
| UBIL | mg/dL | 0.0–1.1 | mg/dl | 0.1–1 | 0 (0%) |
| CBIL | mg/dL | 0.0–0.3 | mg/dl | 0–0.6 | 60 (22%) |
| AST | U/L | 17–59 | U/L | 20–43 | 2 (1%) |
| ALT | U/L | 0–50 | U/L | 11–53 | 10 (4%) |
| AST/ALT* | Ratio | - | ratio | 0.67–2.08 | 272 (100%) |
| ALP | U/L | 38–126 | U/L | 42–135 | 10 (4%) |

Abbreviation: **N**: Sample size, **TP**: Total Protein, **ALB**: Albumin, **GLOB**: Globulin, **A/G**: Albumin/Globulin Ratio, **TBIL**: Total Bilirubin, **UBIL**: Unconjugated (indirect) Bilirubin, **CBIL**: Conjugated (direct) bilirubin, **AST**: Aspartate transaminase, **ALT**: Alanine transaminase, **ALP**: Alkaline phosphatase, **MTH**: Manipal Teaching Hospital, **OOR**: Out of Range, * This parameter is not currently reported in the author's laboratory.

platform. Since our laboratory is part of a tertiary care teaching hospital located at Pokhara and provides diagnostic services to the general population of the entire Gandaki Province, we chose our reference population from the general population of this province. The impact of these new RIs on the prevention, screening, diagnosis, prognosis, and treatment of liver diseases among the Nepali population will be profound. They provide a more accurate framework for interpreting LFT results in the context of the local population. Furthermore, they hold great potential for future biomedical research or clinical trials conducted in Gandaki province.

Since there is still a scarcity of published kinds of literature on dry-chemistry-based RIs for clinical parameters, we had little opportunity to compare and contrast our RIs with others. Even a comparison of our data with wet-chemistry-based external RIs could not be done sufficiently because of the different study designs, the age range of the subjects, analytical methods, and statistical approaches used in their establishment. For these reasons, we could compare our new RIs only with those provided in the OCD Vitros kit inserts. We found that the difference was the biggest in the RIs of AST/ALT ratio (39.6%) followed by A/G ratio (35%), globulin (30%), total protein (29%), and conjugated bilirubin (22%). For others, the difference was minimal ($\leq$7%). Since some of the LFTs such as albumin, A/G ratio and all three fractions of serum bilirubin showed significant association with the sex of the study participants, we also established the sex-based RIs besides for the total population. As expected, the RIs of serum albumin, globulin, A/G ratio, total bilirubin, unconjugated bilirubin, and conjugated bilirubin levels differed significantly between males and females, suggesting the fact that sex-specific RIs are also essential for these tests besides total RIs while interpreting their test results. Though serum total protein, albumin, A/G ratio, bilirubin and ALP levels showed significant association with the age of participants, we were not able to establish a robust and reliable age-group specific RIs for them, as we could not achieve the minimum sample size required for each age group recommended by IFCC-CLSI guidelines. The observed discrepancies between our newly established RIs and the previously adopted ones suggest that a certain percentage of individuals tested for liver functions could be inaccurately diagnosed as diseased according to the adopted RIs, potentially leading to unnecessary treatment.

Studies conducted in various parts of India and China have reported similar upper values for ALT, AST, and ALP, and comparable RIs for serum albumin, total protein, AST, and ALT

[19–21]. However, the RI for ALP differed in the Chinese population [22]. Separate studies on the Canadian reference population have reported sex-specific RIs for ALT and ALP that diverge significantly from our RIs [23]. Sex hormones may contribute to these sex-specific variations in certain parameters such as total and unconjugated bilirubin [24]. Sex-specific variations in the RIs of these LFTs have also been noted in the Ugandian and Rawandian adult population [25, 26].

These discrepancies in RIs between different research findings could be attributed to variations in food and environmental factors, as well as differences in the age and gender of the study groups [27]. It's evident that factors such as diet, physical activity level, genetic makeup, and social circumstances can all influence physiological processes [28, 29]. These variations in RIs among people of different dietary habits, race, ethnicity and geography for LFTs thus underscores the necessity of establishing RIs specific to their own population being served.

## Strength and limitations

Our study followed the latest IFCC-CLSI guidelines to generate the first dry chemistry-based RIs of LFTs for the healthy adult Nepali population under strict quality assurance measures. The enrolled study participants hailed from all 11 districts of Gandaki Province and represented various geographic settings, socioeconomic strata, wider age groups, and gender along with different dietary, drinking and smoking habits. As a result, our new RIs are highly accurate, precise, and trustworthy, and can be used safely to replace the existing imported RIs. Besides, bringing preciseness in disease diagnosis and clinical decision-making processes, they can be used by future biomedical research or clinical trials in this province and exported to the laboratories of neighbouring provinces of Nepal that have not yet established their own RIs. Our study is not free from limitations. Ideally, we should have a thorough clinical and laboratory screening of each study participant before their enrollment, we mostly depended on their self-reported information and recent past medical records. Hence, a few participants that should have been excluded due to the presence of some sort of subclinical diseases might have been included in the study and thus contributed to the bias in our RIs. Since our RIs are specific to the OCD Vitros 4600 dry chemistry platform and the local adult population of Gandaki Province, they may not be applicable as such to those laboratories using different analytical platforms and/or populations outside the Gandaki province of Nepal.

## Conclusion

This study reports the first dry-chemistry-based RIs of 11 different LFTs using a priori-based non-parametric approach for the adult population of Gandaki Province of Nepal. Although comparable to some extent, new RIs for some parameters were quite divergent from the currently adopted external RIs. We also found that gender had a significant influence on the RIs of albumin, globulin, A/G ratio, AST, ALT, and bilirubin levels. These new RIs are expected to displace the adopted RIs and bring precision in the result interpretation, clinical decision making and follow-up the treatment of liver diseases. These new RIs will provide the baseline data for generating pooled national RIs for LFTs, bring harmony in their test result interpretation, and relieve dry chemistry-based laboratories of Gandaki Province from their burden of adopting external RIs.

## Supporting information

**S1 Table. List of chemicals and kit lot used in the study.**
(XLSX)

**S2 Table. Quality control descriptive for liver function test parameters.**
(XLSX)

**S3 Table. Distribution pattern of study participants by their place of residence.**
(XLSX)

**S4 Table. Tests of normality for baseline and LFT parameters.**
(XLSX)

## Acknowledgments

The authors would like to express their gratitude to Manipal Teaching Hospital and Pokhara University, Pokhara for supporting this research. Special thanks and gratitude also go to the entire laboratory staff and faculty members of the Department of Clinical Biochemistry, Manipal Teaching Hospital, Pokhara, Nepal, for their constant assistance in the blood sample collection, processing, and laboratory analyses.

## Author Contributions

**Conceptualization:** Asmita Sharma, Daya Ram Pokharel.

**Funding acquisition:** Asmita Sharma, Ganesh Dhakal.

**Investigation:** Asmita Sharma.

**Methodology:** Asmita Sharma.

**Project administration:** Asmita Sharma.

**Resources:** Daya Ram Pokharel, Ganesh Dhakal.

**Supervision:** Daya Ram Pokharel, Ganesh Dhakal.

**Validation:** Asmita Sharma.

**Visualization:** Asmita Sharma.

**Writing – original draft:** Asmita Sharma.

**Writing – review & editing:** Asmita Sharma.

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
