## [Decision Letter · Decision Letter 0]

18 Aug 2023

PGPH-D-23-00259

Establishment of dry-chemistry-based reference intervals of routine liver function tests for the adult Nepalese population by a priori method

Dear Dr. Sharma,

Thank you for submitting your manuscript to PLOS Global Public Health. After careful consideration, we feel that it has merit but does not fully meet PLOS Global Public Health’s publication criteria as it currently stands. Therefore, we invite you to submit a revised version of the manuscript that addresses the points raised during the review process.

Please note that we have only been able to secure a single reviewer to assess your manuscript. We are issuing a decision on your manuscript at this point to prevent further delays in the evaluation of your manuscript. Please be aware that the editor who handles your revised manuscript might find it necessary to invite additional reviewers to assess this work once the revised manuscript is submitted. However, we will aim to proceed on the basis of this single review if possible.

The reviewer has commented in particular on the study design and communication; please ensure you address each of the reviewer's comments when revising your manuscript.

We look forward to receiving your revised manuscript.

Kind regards,

Hugh Cowley

Staff Editor

Journal Requirements:

1. We ask that a manuscript source file is provided at Revision. Please upload your manuscript file as a .doc, .docx, .rtf or .tex.

2. We do not publish any copyright or trademark symbols that usually accompany proprietary names, eg  ©, ®, ™  (e.g. next to drug or reagent names). Please remove all instances of trademark/copyright symbols throughout the text, including ® on pages 7 & 8.

4. We notice that your supplementary files are included in the manuscript file. Please remove them and upload them with the file type 'Supporting Information'. Please ensure that each Supporting Information file has a legend listed in the manuscript after the references list.

5. In the online submission form, you indicated that "The datasets generated and analyzed during the current study would be available from the corresponding author on personal request". All PLOS journals now require all data underlying the findings described in their manuscript to be freely available to other researchers, either 1. In a public repository, 2. Within the manuscript itself, or 3. Uploaded as supplementary information.

Additional Editor Comments (if provided):

Reviewers' comments:

Reviewer's Responses to Questions

**Comments to the Author**

1. Does this manuscript meet PLOS Global Public Health’s publication criteria? Is the manuscript technically sound, and do the data support the conclusions? The manuscript must describe methodologically and ethically rigorous research with conclusions that are appropriately drawn based on the data presented.

Reviewer #1: Yes

2. Has the statistical analysis been performed appropriately and rigorously?

Reviewer #1: Yes

3. Have the authors made all data underlying the findings in their manuscript fully available (please refer to the Data Availability Statement at the start of the manuscript PDF file)?

Reviewer #1: Yes

4. Is the manuscript presented in an intelligible fashion and written in standard English?

Reviewer #1: Yes

5. Review Comments to the Author

Reviewer #1: The authors have conducted a cross-sectional study to establish population specific reference intervals for liver function tests in Nepal. The methodology adopted and statistical methods used are very robust and relevant.

1. The authors need to justify Why population specific references are needed especially for Nepalese population. If everyone establishes their population specific intervals, the entire medical research must be repeated in every population which is not feasible. Where is the end to this? The authors need to comment on this and discuss.

2. The participants are considered as healthy based on the questionnaire and self-declaration which was also mentioned in the limitations. This was acceptable if the derived reference ranges were similar to the global ranges. Since the proposed ranges for this population are different, the participants should have been clinically tested and determined healthy. Attest those participants who are determined as “out of range” must be clinically examined and declared as healthy if these values have to be acceptable.

3. Since the authors are proposing the derived values in this study as references for the entire Nepalese population, they should have participants from more than one center across the country.

4. At several places in the manuscript, the references are giving an error “Error! Reference source not found” This has to be rectified.

6. PLOS authors have the option to publish the peer review history of their article (what does this mean?). If published, this will include your full peer review and any attached files.

**Do you want your identity to be public for this peer review?** For information about this choice, including consent withdrawal, please see our Privacy Policy.

Reviewer #1: No

---

## [Decision Letter · Decision Letter 1]

18 Jan 2024

PGPH-D-23-00259R1

Establishment of dry-chemistry-based reference intervals of routine liver function tests for the adult population of Gandaki Province, Nepal.

Dear Dr. Sharma,

Thank you for submitting your manuscript to PLOS Global Public Health. After careful consideration, we feel that it has merit but does not fully meet PLOS Global Public Health’s publication criteria as it currently stands. Therefore, we invite you to submit a revised version of the manuscript that addresses the points raised during the review process.

Your revision is reviewed by a new reviewer. Please provide your responses to the comments point-by-point.

We look forward to receiving your revised manuscript.

Kind regards,

Jianhong Zhou

Staff Editor

Journal Requirements:

2. Some material included in your submission may be copyrighted. According to PLOS’s copyright policy, authors who use figures or other material (e.g., graphics, clipart, maps) from another author or copyright holder must demonstrate or obtain permission to publish this material under the Creative Commons Attribution 4.0 International (CC BY 4.0) License used by PLOS journals. Please closely review the details of PLOS’s copyright requirements here: PLOS Licenses and Copyright. If you need to request permissions from a copyright holder, you may use PLOS's Copyright Content Permission form.

Potential Copyright Issues:

Fig 1: please (a) provide a direct link to the base layer of the map (i.e., the country or region border shape) and ensure this is also included in the figure legend; and (b) provide a link to the terms of use / license information for the base layer image or shapefile. We cannot publish proprietary or copyrighted maps (e.g. Google Maps, Mapquest) and the terms of use for your map base layer must be compatible with our CC-BY 4.0 license. 

"

Additional Editor Comments (if provided): We note that one or more reviewers has recommended that you cite specific previously published works. As always, we recommend that you please review and evaluate the requested works to determine whether they are relevant and should be cited. It is not a requirement to cite these works. We appreciate your attention to this request.

Reviewers' comments:

Reviewer's Responses to Questions

**Comments to the Author**

1. If the authors have adequately addressed your comments raised in a previous round of review and you feel that this manuscript is now acceptable for publication, you may indicate that here to bypass the “Comments to the Author” section, enter your conflict of interest statement in the “Confidential to Editor” section, and submit your "Accept" recommendation.

Reviewer #2: (No Response)

2. Does this manuscript meet PLOS Global Public Health’s publication criteria? Is the manuscript technically sound, and do the data support the conclusions? The manuscript must describe methodologically and ethically rigorous research with conclusions that are appropriately drawn based on the data presented.

Reviewer #2: Partly

3. Has the statistical analysis been performed appropriately and rigorously?

Reviewer #2: Yes

4. Have the authors made all data underlying the findings in their manuscript fully available (please refer to the Data Availability Statement at the start of the manuscript PDF file)?

Reviewer #2: Yes

5. Is the manuscript presented in an intelligible fashion and written in standard English?

Reviewer #2: Yes

6. Review Comments to the Author

Reviewer #2: 28 December 2023

Ms. Ref. No.: PGPH-D-23-00259R1

Journal: PLOS Global Public Health.

Title: Establishment of dry-chemistry-based reference intervals of routine liver function tests for the adult population of Gandaki Province, Nepal.

Comments:

Thank you for your efforts in writing this article (Research Article) on a very pertinent topic. I have some observations where mentioned in the following paragraphs that will be useful for its improvement:

1- The research consists of 11 parameters commonly used in LFTs. Why selecting only these 11 factors?

2- The normal weight for BMI is about 18.5 to 24.9 while in this study it is between18.5-27.5. Please reintroduce this selected rang.

3- Rang of age for participants is 18-70 years, what is the main reason for that?

4- What about the association of BMI and age with this result?

5- Participants were checked for a set of medical history-centered questionnaires, general physical examination, and certain screening blood tests which made general screening, while there is some question in following lines.

• The fatty liver is a main factor in this screening or not?

• Which method/methods was/were used for diagnosing of fatty liver?

• Has the life style of participants any association with results?

• What about who used any kinds of alcohol? Were they checked by questionnaire? And did they leave the study or not?

6- It seems to be suitable adding an inclusion and exclusion chart to manuscript for this study.

7- Moreover, The Following reference can be included in the introduction part for more readability:

• Age-specific reference intervals for liver function tests in healthy neonates, infants, and young children in Iran. https://doi.org/10.1002/jcla.24995

• Age-specific reference intervals for routine biochemical parameters in healthy neonates, infants, and young children in Iran. https://doi.org/10.1111/jcmm.17646

• Reference intervals for routine biochemical markers and body mass index: A study based on healthcare center database in northeastern Iran. https://doi.org/10.1002/iub.2437

• Pediatric reference intervals for hematology parameters in healthy infants and young children in Iran. https://doi.org/10.1111/ijlh.14132

• Interaction between the genetic variant of rs696217-ghrelin and food intake and obesity and dyslipidemia. https://doi.org/10.1111/ahg.12443

7. PLOS authors have the option to publish the peer review history of their article (what does this mean?). If published, this will include your full peer review and any attached files.

**Do you want your identity to be public for this peer review?** For information about this choice, including consent withdrawal, please see our Privacy Policy.

Reviewer #2: **Yes: **Reza Assaran

---

## [Decision Letter · Decision Letter 2]

8 Apr 2024

Establishment of dry-chemistry-based reference intervals of routine liver function tests for the adult population of Gandaki Province, Nepal.

PGPH-D-23-00259R2

Dear Mrs Sharma,

We are pleased to inform you that your manuscript 'Establishment of dry-chemistry-based reference intervals of routine liver function tests for the adult population of Gandaki Province, Nepal.' has been provisionally accepted for publication in PLOS Global Public Health.

Best regards,

Julia Robinson

Executive Editor

Reviewer Comments (if any, and for reference):

Reviewer's Responses to Questions

**Comments to the Author**

1. If the authors have adequately addressed your comments raised in a previous round of review and you feel that this manuscript is now acceptable for publication, you may indicate that here to bypass the “Comments to the Author” section, enter your conflict of interest statement in the “Confidential to Editor” section, and submit your "Accept" recommendation.

Reviewer #2: All comments have been addressed

2. Does this manuscript meet PLOS Global Public Health’s publication criteria? Is the manuscript technically sound, and do the data support the conclusions? The manuscript must describe methodologically and ethically rigorous research with conclusions that are appropriately drawn based on the data presented.

Reviewer #2: Partly

3. Has the statistical analysis been performed appropriately and rigorously?

Reviewer #2: Yes

4. Have the authors made all data underlying the findings in their manuscript fully available (please refer to the Data Availability Statement at the start of the manuscript PDF file)?

Reviewer #2: Yes

5. Is the manuscript presented in an intelligible fashion and written in standard English?

Reviewer #2: Yes

6. Review Comments to the Author

Reviewer #2: Thanks

7. PLOS authors have the option to publish the peer review history of their article (what does this mean?). If published, this will include your full peer review and any attached files.

**Do you want your identity to be public for this peer review?** For information about this choice, including consent withdrawal, please see our Privacy Policy.

Reviewer #2: **Yes: **Reza Assaran
